# Development of a Diagnostic Microfluidic Chip for SARS-CoV-2 Detection in Saliva and Nasopharyngeal Samples

**DOI:** 10.3390/v16081190

**Published:** 2024-07-25

**Authors:** Sandhya Sharma, Massimo Caputi, Waseem Asghar

**Affiliations:** 1Department of Electrical Engineering and Computer Science, Florida Atlantic University, Boca Raton, FL 33431, USA; 2Asghar-Lab, Micro and Nanotechnology in Medicine, College of Engineering and Computer Science, Boca Raton, FL 33431, USA; 3Charles E. Schmidt College of Medicine, Florida Atlantic University, Boca Raton, FL 33431, USA; mcaputi@health.fau.edu; 4Department of Biological Sciences (Courtesy Appointment), Florida Atlantic University, Boca Raton, FL 33431, USA

**Keywords:** SARS-CoV-2 detection, Point-of-care diagnostics, RT-LAMP-based microfluidic chip

## Abstract

The novel coronavirus SARS-CoV-2 was first isolated in late 2019; it has spread to all continents, infected over 700 million people, and caused over 7 million deaths worldwide to date. The high transmissibility of the virus and the emergence of novel strains with altered pathogenicity and potential resistance to therapeutics and vaccines are major challenges in the study and treatment of the virus. Ongoing screening efforts aim to identify new cases to monitor the spread of the virus and help determine the danger connected to the emergence of new variants. Given its sensitivity and specificity, nucleic acid amplification tests (NAATs) such as RT-qPCR are the gold standard for SARS-CoV-2 detection. However, due to high costs, complexity, and unavailability in low-resource and point-of-care (POC) settings, the available RT-qPCR assays cannot match global testing demands. An alternative NAAT, RT-LAMP-based SARS-CoV-2 detection offers scalable, low-cost, and rapid testing capabilities. We have developed an automated RT-LAMP-based microfluidic chip that combines the RNA isolation, purification, and amplification steps on the same device and enables the visual detection of SARS-CoV-2 within 40 min from saliva and nasopharyngeal samples. The entire assay is executed inside a uniquely designed, inexpensive disposable microfluidic chip, where assay components and reagents have been optimized to provide precise and qualitative results and can be effectively deployed in POC settings. Furthermore, this technology could be easily adapted for other novel emerging viruses.

## 1. Introduction

The severe acute respiratory syndrome coronavirus 2 (SARS-CoV-2) infection is the cause of the pandemic of acute respiratory disease, named ‘coronavirus disease 2019’ (COVID-19). The disease was first reported in Wuhan, China, in December 2019 and was declared a worldwide pandemic by the World Health Organization on 11 March 2020 [1,2]. SARS-CoV-2 is a positive- sense single-stranded RNA virus closely related to the coronaviruses that caused the outbreaks of severe acute respiratory syndrome (SARS) in the year 2002–03 and Middle East respiratory syndrome (MERS) in the year 2012 [3,4,5]. A worldwide vaccination campaign is ongoing and over 70% of the world population has received at least one vaccine dose. However, the chance of virus transmission remains high due to the high risk of re-infection and the emergence of new variants that might differ in infectivity and pathogenicity [6]. Currently, the prevalent circulating viruses in the US are variants of the original Omicron strain (BA.1), which is no longer circulating [7]. The availability of rapid and easy-to-use diagnostics can play a significant role in limiting the spread of the virus and in the detection of novel variants of concern. Lateral flow and other antigenic assays are the most widely used methods for the detection of SARS-CoV-2. These assays are widely available, can be self-administered, are cheap to fabricate, and can be easily stored. Nevertheless, the majority of the rapid antigen tests utilized have a high false negative testing ratio and their limits of detection (LoDs) are over 10^5^–10^6^ viral genome copies/mL [8]. Although antigenic tests provide an excellent approach to the rapid detection of the virus [9,10], they can seldom identify the virus during the early phase of the infection and they lack the sensitivity required for a reliable clinical diagnosis. Since individuals with little to no symptoms and a lower viral load can carry and spread the virus, false negative results might greatly impact public health policies, especially when a novel, more transmissible variant with unknown pathogenic potential emerges [11,12,13,14].

Nucleic acid amplification tests (NAATs) based on the reverse transcriptase quantitative polymerase chain reaction (RT-qPCR) are the most accurate and reliable methods for the detection of SARS-CoV-2 and are the gold standard for viral diagnose s [15,16,17]. RT-qPCR requires specialized equipment and highly trained technicians for sample processing and longer turnover time to results, making it a less desirable solution in p oint-of-c are (POC) testing, where a quick turnaround is required and specialized equipment and personnel are usually not available. To date, the FDA has approved more than 23 POC devices under Emergency Use Authorization (EUA). About 12 of these devices utilize an RT-qPCR assay, while 11 utilize RT-loop-mediated isothermal amplification (RT-LAMP) [18], which allows for the amplification of the target nucleic acid at a constant temperature, alleviating the need for a thermocycler to be incorporated in the testing device [19]. However, the majority of the POC devices (e.g., Abbott diagnostics ID Now, Cepheid Xpert, and Roche Cobas) often exhibit low sensitivity and a high rate of false negative results coupled with expensive equipment and a high price target per test [20,21]. Thus, the molecular diagnostic tests for SARS-CoV-2 currently available for POC settings remain inefficient in meeting global demand, especially in resource-constrained locations [22].

Most of the molecular tests on the market collect nasopharyngeal (NP) and oropharyngeal swab samples that may cause uneasiness in patients and can be difficult to self-administer. Conversely, saliva can be easily self-collected, reducing the interaction between the patient and healthcare workers, and can be reliably utilized as a source for the detection of coronavirus and other respiratory viruses [23,24,25,26,27,28]. Researchers have developed multiple saliva-based methods and protocols; nevertheless, only 1 in 10 of the EUA-approved NAAT protocols and diagnostic systems utilize saliva and these methods require trained personnel and expensive equipment for target amplification and detection [29,30,31,32,33,34]. An automated, affordable, and rapid molecular saliva-based test has yet to be developed.

Herein, we describe a novel RT-LAMP-based microfluidic chip and set-up that offer a reduced workflow, thereby reducing the time of diagnosis (Figure 1). The automated diagnostic set-up processes a saliva sample to detect the SARS-CoV-2 virus within 40 min. The microfluidic chip utilizes magnetic beads for the isolation and purification of the viral RNA from the saliva sample and uses RT-LAMP to amplify the target. The SYBR Green 1 dye, which changes color from orange to green on target amplification, is used to enable the visual detection of the amplified target sequence. The RT-LAMP-based microfluidic chip exhibited a limit of detection (LoD) of 3 × 10^4^ viral RNA copies/reaction (5.8 × 10^5^ copies/mL) from saliva samples within 40 min. The analytical sensitivity (or LoD) of the devices increased to ~40 virion copies/reaction (580 copies/mL) when nasopharyngeal samples were tested. The device we developed can efficiently process diverse samples for rapid SARS-CoV-2 detection in POC settings, providing early diagnose s and help in the monitoring and control of the disease’s spread.

## 2. Materials and Methods

### 2.1. RT-LAMP Primer Design and Testing

A total of 75 r andom SARS-CoV-2 genomic variant sequences (from the National Center for Biotechnology Information (NCBI)) were analyzed utilizing the sequence alignment tool Clustal Omega to determine a highly conserved target region for the design of the RT-LAMP primers. Primers were designed utilizing the Primer Explorer V5 (Eiken Chemical Co. Ltd., Tokyo, Japan) and synthesized by Integrated DNA Technologies (IDT). The SARS-CoV-2 nucleocapsid plasmid control (IDT # 10006625) was utilized as a positive LAMP amplification control. The heat-inactivated purified virions of SARS-CoV-2 (BEI Resources, Manassas, VA, USA, NR-52286) and the genomic RNA (BEI Resources, NR-52347) from isolate USA-WA1/2020 were utilized as positive controls for RNA extraction and RT-LAMP. The purified genomic RNA (BEI resources, NR-52349) and the purified inactivated virions of SARS-COV-1, Urbani strain (BEI resources, NR-52346), were utilized as negative controls.

Saliva and nasopharyngeal samples from de-identified healthy donors were collected 1–2 h before testing. The samples were purposely spiked with either different amounts of SARS-CoV-2 purified RNA or inactivated virus. Nucleic acids were isolated from the samples using the Dynabeads SILANE viral Nucleic Acid (NA) kit (Invitrogen, Carlsbad, CA, USA). In total, 100 μL of saliva or the nasopharyngeal swab was incubated in 300 μL of a lysis/binding buffer for 5 min. Samples were incubated with the magnetic Dynabeads to isolate the nucleic acids (RNA or DNA), washed, and eluted in 50 μL of the elution buffer. The LavaLAMP RNA Master Mix kit (Lucigen, Middleton, WI, USA) was utilized in all LAMP reactions. A 20 μL RT-LAMP reaction was set up with Master Mix (12.5 μL), RT-LAMP primers (2.5 μL), and the target negative/positive control (10 μL). The AriaMx Real-Time PCR system (Agilent, Santa Clara, CA, USA) was utilized to maintain 70 °C (LavaLAMP recommended temperature range: 68–74 °C) for 40 min for the RT-LAMP isothermal amplification in benchtop experiments. In total, 1 μL of SYBR Green I nucleic A dye (Invitrogen) was added to the reaction solution after the isothermal incubation for the colorimetric analysis.

**RT-qPCR reaction**: The PCR primers and TaqMan probe for the qPCR test were based on the N1 and N2 regions of SARS-CoV-2. Each qPCR reaction mix contained 10 μL of SARS-CoV-2 primers and TaqMan probe mix, 10 μL of RNA Master Mix, and 5 μL of the eluted target. RT-qPCR was carried out with a Thermal Cycler (AriaMx Real-Time PCR system) with a temperature profile of reverse transcription (RT) at 50 °C for 15 min and denaturation at 95 °C for 1 min, followed by 45 cycles of amplification (95 °C for 15 s and 60 °C for 30 s).

### 2.2. Design and Fabrication of Microfluidic Chips

Microfluidic chips were fabricated using Poly(methyl methacrylate) (PMMA) sheets. The top and base layers are 750 μm thick and the inner well layer is 1.5 mm thick. The PMMA layers were attached using double-sided adhesive (DSA) tape (Appendix A). A CO_2_ laser cutter was used to fabricate the chambers as previously reported [34,35,36,37,38,39,40,41]. Each microfluidic chip contains one diamond-shaped sample inlet chamber, two washing buffer chambers, one reaction chamber detached by three elliptical-shaped valving chambers containing mineral oil (viscosity—15 cSt), and one unconnected oval-shaped sensor chamber next to the reaction chamber for the temperature sensor attachment (Figure 2a shows the fully assembled chip and chambers, illustrating their functions). Microfluidic chips were UV-irradiated for 30 min to sterilize them. All the inlets were then blocked with scotch tape. The scotch tape was removed from the microfluidic chip for the filling of reagents before sample testing.

### 2.3. Magnetic Actuation of Diagnostic Platform Set-Up

The automated diagnostic platform (shown in Figure 2b) had been previously designed and optimized by our lab [42,43]. The diagnostic platform controls the movement of the magnetic beads, which are guided by two small magnets (5 mm diameter neodymium) located in a 3D- printed inclusion in the platform. The magnets move bidirectionally, and their movement is synchronized by a stepper motor. The stepper motor is directed by a printed circuit board (PCB) integrated with Arduino. The movement of the magnetic beads from one chamber to another and the incubation time is controlled by a G-code scripted in Python. We utilized an Arduino-based temperature controller to maintain the 70 °C temperature required for on-chip amplification [40]. The k-type sensor probe (dia—0.13 mm) is used to read the actual temperature of the reagent in real-time. The Arduino shield powers the 1.98 watts 2 cm × 2 cm rectangular-shaped ultra-thin nano carbon flexible heater (TSA(C) 0200020eR12.6, Pelonis Tech Exton, PA, USA) by controlling the MOSFET gate signals; therefore, no external power source is required to operate the temperature controller system. The sensor probe is placed in the sensor chamber and the ultra-thin nano- heater is placed on the surface of the sensor and reaction chamber.

### 2.4. On-Chip Detection of SARS-CoV-2 from Saliva and Nasopharyngeal Samples

The sample inlet chamber was loaded with magnetic beads + isopropanol (Dynabeads SILANE viral Nucleic Acid (NA) KIT). Washing buffers, I and II (Dynabeads SILANE viral Nucleic Acid (NA) KIT), were added to the washing chambers, and their viscosity was increased by adding RNase/DNase-free water in a 1:1 ratio. LavaLAMP RNA Master Mix (Lucigen), SARS-CoV-2- specific primers, and elution buffer were added to the reaction chamber and the sensor chamber solution, respectively. It takes less than 5 min to manually fill the reagents into the chip. The MgSO_4_ concentration of LavaLAMP RNA Master Mix was increased from 5 mM to 9.8 mM for on-chip testing. Lysis of the virus in saliva samples was carried out in an Eppendorf tube. In total, 100 μL of the saliva sample was added to a 300 μL lysis/binding buffer (Dynabeads SILANE viral Nucleic Acid (NA) KIT) and incubated on a rocker for 5 min. Following this, 200 μL from the 400 μL solution (100 μL of the saliva + 300 μL lysis/binding buffer) was loaded into the inlet chamber of the pre-filled chip and placed on the automated diagnostic platform. For nasopharyngeal samples, the swab stick spiked with 10 μL of a virus sample was incubated in a 300 μL lysis/binding buffer for 5 min. Following this, 200 μL from the 300 μL solution containing the lysis/binding buffer and the spiked virus was loaded in the inlet chamber of the pre-filled chip and placed on the automated diagnostic platform. Next, the heater and magnetic actuator were started concurrently. The RNA released from the lysed virions was bound to magnetic beads. The beads were moved to the washing chambers (1 min 30 s in each chamber), and then to the reaction chamber for the elution of the RNA, reverse transcription, and amplification. The temperature of the chip was maintained at 70 °C for 40 min for the LAMP amplification reaction. After 40 min of amplification, the heater was removed from the microfluidic chip, and SYBR Green 1 dye was added right away for the colorimetric analysis. All the on-chip experiments were repeated a minimum of three times. The on-chip images were analyzed with ImageJ 1.53i (National Institutes of Health) to measure the saturated gray intensity value to validate the naked-eye results.

## 3. Results and Discussion

### 3.1. Benchtop RT-LAMP Amplification Results of Spiked Samples

Several RT-LAMP-based microfluidic diagnostics have been developed for various diseases [12,13,14,21,22,23,29,30,31,32,40,43]. In this research, we have developed a promising approach that offers a naked-eye result analysis and automated RNA extraction from the self-collected saliva samples at a low cost. For the initial primer designing, we analyzed the sequence of 75 SARS-CoV-2 isolates (Appendix A) and identified a highly conserved sequence within the nucleocapsid phosphoprotein (N) gene, which was highly specific for SARS-CoV-2. Appendix A illustrate ClustalW and NCBI BLAST results providing evidence that the chosen target sequence is a conserved sequence among the 75 sequences and exclusive for other c oronaviruses found in humans. This conserved sequence was used to design SARS-CoV-2 LAMP primers (Figure 3a,b). The analytical sensitivity and specificity of the RT-LAMP primers were tested using different concentrations of SARS-CoV-2 RNA whereas SARS-COV RNA was used as a negative control. The bench-top (off-chip) assays were performed in a thermocycler and the results were visualized by a colorimetric assay that utilized the SYBR Green 1 dye, which changes color from orange to green in the presence of double-stranded DNA (Figure 4a). The lowest limit of detection observed was 4 RNA copies/reaction, whereas negative controls showed no amplification, validating that the design primers would bind only to SARS-CoV-2 targets. In agreement with these results, gel electrophoresis of the reaction products showed the presence of amplicons only in the reactions containing the SARS-CoV-2 template (Appendix A). Next, we tested the accuracy of the primers and the efficiency of the RT-LAMP reaction when the starting sample was saliva- spiked with plasmid DNA containing the viral target sequence. We utilized magnetic Dynabeads to extract the plasmid DNA from the saliva and we were able to reliably detect as low as 70 copies (3.5 × 10^3^ copies/mL) of the target sequence (Figure 4b and Appendix A). These results validate that the magnetic beads are capable of extracting the target from the saliva samples. Saliva is difficult to pipette due to its thickness and stringiness. The lysis buffer from the kit reduced the viscosity of the saliva, making it compatible with the microfluidic chip to extract the target. Following this, the saliva samples were spiked with inactivated SARS-CoV-2 virions. The viral RNA was extracted utilizing magnetic Dynabeads and analyzed by RT-LAMP in a thermocycler. This assay had a limit of detection (LoD) of roughly 1 × 10^3^ viral copies/reaction (Figure 4c). This implies that the viral NA kit prevented RNA degradation and facilitated optimal RNA extraction from a small saliva volume. RT-qPCR amplification further validates the optimal viral RNA isolation from the saliva samples utilizing magnetic Dynabeads, achieving an LoD of ~60 viral RNA copies/reaction (Figure 5). RT-qPCR provided greater sensitivity compared to the presented method; however, the conventional RT-qPCR method involves multiple lab-based manual steps and is time-consuming. The manufacturer initially designed the Dynabeads™ SILANE viral NA kit to process human serum/plasma samples. RT-qPCR results show that the kit is efficient in RNA extraction not only from serum/plasma but also from saliva samples. The discrepancy in the amplification curves could be attributed to RNA degradation during the manual processing of the thick and stringent saliva samples.

### 3.2. Microfluidic Chip and Automated Platform

To implement viral testing for a greater number of people with a substantially reduced cost, we designed a microfluidic chip compiling different processes on a single platform for saliva/nasopharyngeal samples. The microfluidic chip contains multiple chambers for the sample inlet, RNA isolation, washing, elution, and amplification (Figure 2a). All reagents were pre-loaded into the respective chambers of the microfluidic chip before running the assay. The reaction chamber contains RT-LAMP primers, the reaction master mix, and the elution buffer. Since PMMA adsorbs polar molecules on its surface and hinders on-chip amplification, the recommended RT-LAMP reaction master mix was modified by increasing the MgSO_4_ concentration from 5 mM to 9.8 mM. The addition of Mg^++^ ions improves the polymerase activity and stabilizes the DNA on the PMMA surface [44,45]. Once the sample was loaded into the inlet chamber, the viral RNA binds to the Dynabeads surface and magnets located on the actuation platform move the Dynabeads. The Dynabeads were moved through the two washing chambers that contain washing buffers. The washing steps remove compounds that might degrade the RNA or inhibit the RT-LAMP reaction. The beads finally made their way to the reaction chamber, where the RNA was eluted, reverse- transcribed, and amplified at 70 °C. The Dynabeads after the RNA elution were moved back to the washing chamber. The isothermal 70 °C temperature was maintained using a small surface heater and a sensor for 40 min. The amplification was visualized upon the addition of SYBR Green 1 dye, which changes in color from orange to green/yellow in the presence of double-stranded DNA amplification products. This magnetic-based sample processing set-up saves technicians time by removing manual steps and helping reduce overall processing time. The workflow in the developed chip is presented on a single platform in such a way that once the sample is loaded, the instrumental set-up carries out the programmed target extraction protocol automatically using magnetic beads. Appendix A shows the workflow of the magnetic beads inside the microfluidic chambers. It is a time-lapse video demonstrating target isolation, washing, and elution in the reaction chamber. Once the target is eluted, the reaction chamber is incubated for 40 min at 70 °C. The results are visible in terms of “yes”—green or “no”—orange after the addition of SYBR Green 1 dye to the reaction chamber.

### 3.3. On-Chip Saliva/Nasopharyngeal Spiked Sample

SARS-CoV-2 viral load is elevated in saliva, ranging between 10^3^ and 10^8^ copies/mL with a median of 10^5^–10^7^ copies/mL [46,47,48]. We processed saliva samples spiked with inactivated SARS-CoV-2 on the chip for 40 min to test the LoD of the microfluidic chip and the automated magnetic actuator set-up when saliva is utilized as a virus source. The automated on-chip assay was able to detect as low as ~3 × 10^4^ viral RNA copies loaded/reaction (Figure 4d). The on-chip resultant images were further analyzed using ImageJ 1.53i (National Institutes of Health) software to measure gray intensity values. The intensity values were obtained from three areas of the image. The average intensity values (minus background) and standard deviation were plotted. To distinguish between negative and positive amplification, we defined 20 arbitrary units [a.u] as a threshold value. Negative amplification reactions showed a lower value than the threshold and positive samples showed significantly higher values. Appendix A represents the gray intensity values of Figure 4d. The defined threshold value (20 arbitrary units [a.u]) clearly shows the difference between negative and positive reactions, validating the naked-eye results. Saliva samples from asymptomatic patients have a median viral load of 10^5^–10^7^ copies/mL, which is similar to the initial viral load observed in symptomatic patients. A week after symptom onset, the SARS-CoV-2 RNA in saliva increases rapidly, from 10^5^ to 10^9^ copies/mL [48]. Given its low LoD, the diagnostic set-up we describe could be extremely useful during the early phase of infection from saliva samples. Since saliva contains proteases and RNases that might increase the degradation rate of the viral RNA during the RNA extraction step and the viral load detected in nasopharyngeal samples is often higher than the one detected in saliva [49,50], we also tested the LoD of our diagnostic set-up utilizing nasopharyngeal swabs as a starting sample. The nasopharyngeal swabs were obtained from healthy donors and spiked with inactivated SARS-CoV-2 virions. In a preliminary assay, the RNA was extracted from nasopharyngeal samples utilizing Dynabeads, and the RT-LAMP reaction was performed in a thermocycler, achieving an LoD of ~10 viral RNA copies/reaction (Figure 6a). Thus, we confirmed the high analytical sensitivity of our primers in an off-chip assay. Next, we processed the nasopharyngeal samples on the chip, obtaining an LoD of ~40 viral RNA copies loaded/reaction (580 copies/mL) (Figure 6b), confirming the high analytical sensitivity of the set-up. Furthermore, the gray intensity graph Appendix A (average values and standard deviation) follows the same trend. On-chip nasopharyngeal SARS-CoV-2 positive sample’s gray scale value is significantly higher than 20 a.u. (threshold value), and negative amplification reactions showed a value lower than the threshold, confirming the colorimetric results.

Since the majority of patients in the early stage of the infection exhibit a viral load between 10^5^ and 10^9^ copies/nasopharyngeal swab sample [47,51], the assay we have described can provide the desired sensitivity to detect most cases in the early stages of infection utilizing nasopharyngeal swab samples. Furthermore, patients carry a saliva viral load between 1 × 10^3^ and 1 × 10^8^ copies/mL [52,53,54] and the proposed assay has an LoD of ~3 × 10^4^ viral RNA copies (5.8 × 10^5^ copies/mL) when the starting material is saliva; therefore, this microfluidic chip can be reliably utilized for the early diagnosis of infection from the majority of saliva samples. We have optimized the beads, buffers, and on-chip protocol for two different types of samples, making this automated system more affordable.

Many researchers have developed operational set-ups for SARS-CoV-2 detection from saliva samples. Appendix A illustrates the limitations of these existing saliva-based rapid molecular tests. Some of the researchers are targeting “extraction-free” methods. However, the main limitation of these methods is the low sensitivity that can be improved with heat treatment plus protease incubations, which require human intervention [55,56,57,58,59,60,61,62]. Additionally, for any manual step involved, trained personnel are required to set up the LAMP reaction. A high throughput from these methods is also feasible, but that requires sophisticated lab settings, making it a limitation for point-of-care settings [57,59,61,63,64,65]. Some of these assays are yet to be developed for POC settings. The diagnostic microfluidic chip we presented uses a small volume of reagents and simplified instruments to detect SARS-CoV-2 from the patient’s saliva or a nasopharyngeal sample with high analytical sensitivity and specificity. The microfluidic chip is expected to cost less than USD 3 for materials, fabrication, and reagents if produced on a large industrial scale. The platform used in this assay is fully programmed and automated and has a reusable battery-operating set-up that can be fabricated for less than USD 50. This 3D- printed platform is made of an electronic circuit integrated with a microcircuit modulated by Arduino to control the movements of the magnets, a step-up motor, and a circuit for the power supply [39]. The listed cost of the material used in the diagnostic platform manufacturing and volume of reagents used in the microfluidic chip are presented in the Appendix A. This detection set-up also avoids carryover contamination in the form of aerosols since the amplicons are retained inside the closed amplification chamber for an endpoint analysis. The primers we selected are mapped on a highly conserved region of the N gene, which is unlikely to change drastically in new emerging variants and strains since it has remained conserved in all strains since the appearance of the virus in late 2019. Appendix A is the SnapGene Viewer results of primers’ binding sites across different variants (Alpha, Beta, Gamma, Delta, and Lambda) of SARS-CoV-2 that emerged over time. These results show evidence that the target site of the primer sequences designed in this research remained conserved. In case of mutations in the primer binding regions due to genetic drift of the viral genome, the primers of the assay can be easily re-designed and adapted. Overall, the system we described could become an important tool for SARS-CoV-2 detection and surveillance and limiting its spread since it can be used for the rapid detection of the viral pathogen at POC settings including doctors’ offices, community health centers, urgent care centers, or remote areas in developing countries cost-effectively.

## 4. Conclusions

Our work describes a disposable RT-LAMP-based microfluidic chip integrated with a portable automated set-up that efficiently captures viral targets for disease detection with reduced workforce, time, components, and reagents. Our results show that this set-up can detect SARS-CoV-2 from the saliva and nasopharyngeal samples using minimal resources. The portable system proposed in this study is fully automated, is powered by an independent source of electricity, does not require user input after the initial loading of the sample, and provides test results in 40 min with good analytical sensitivity to detect SARS-CoV-2 in saliva and nasopharyngeal swab samples.

## Figures and Tables

**Figure 1 viruses-16-01190-f001:**
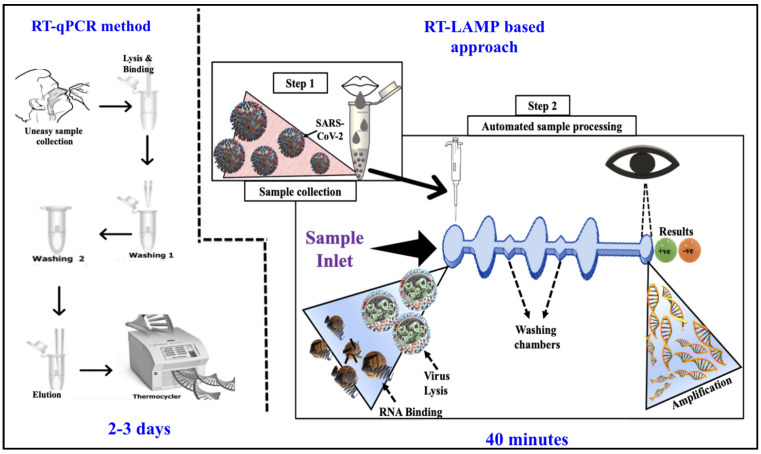
Comparison between the conventional RT-qPCR-based manually processed benchtop testing method (**left**) vs. RT-LAMP-based diagnostic microfluidic chip presented in this paper (**right**).

**Figure 2 viruses-16-01190-f002:**
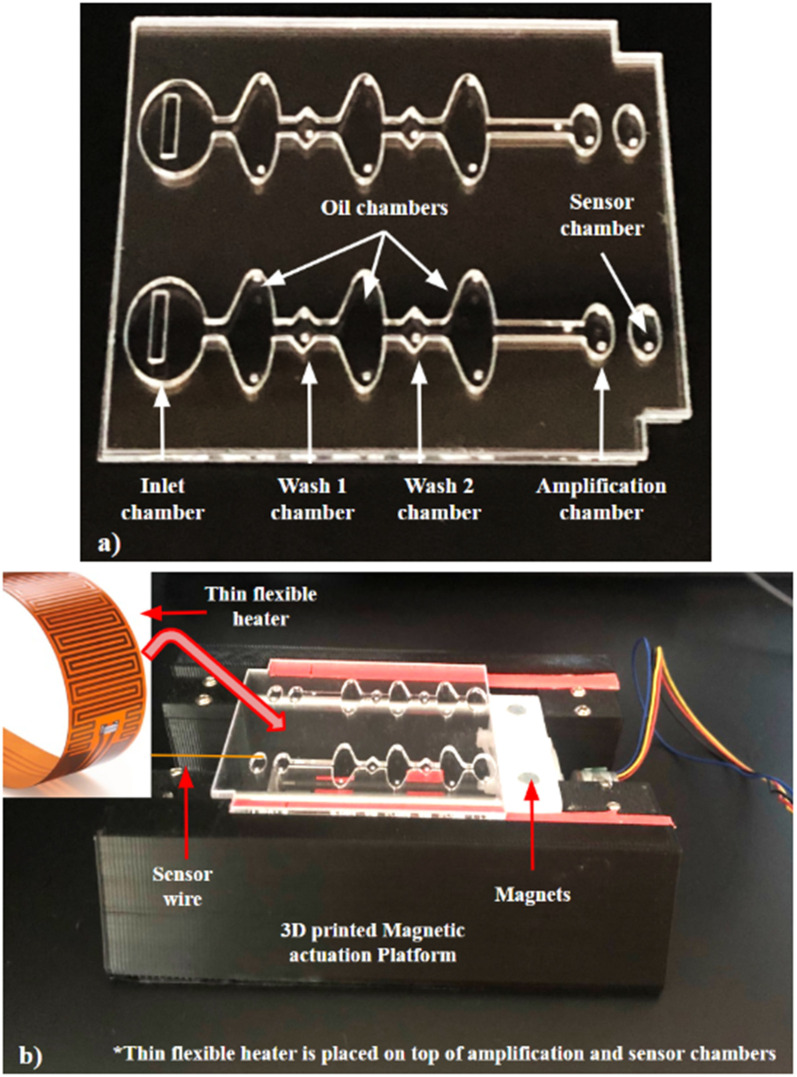
(**a**) Fully assembled microfluidic chip consisting of different chambers for sample loading, RNA isolation, washing, elution, and amplification. (**b**) A utomatic molecular diagnostic set-up with integrated capability of magnetic actuation and heating.

**Figure 3 viruses-16-01190-f003:**
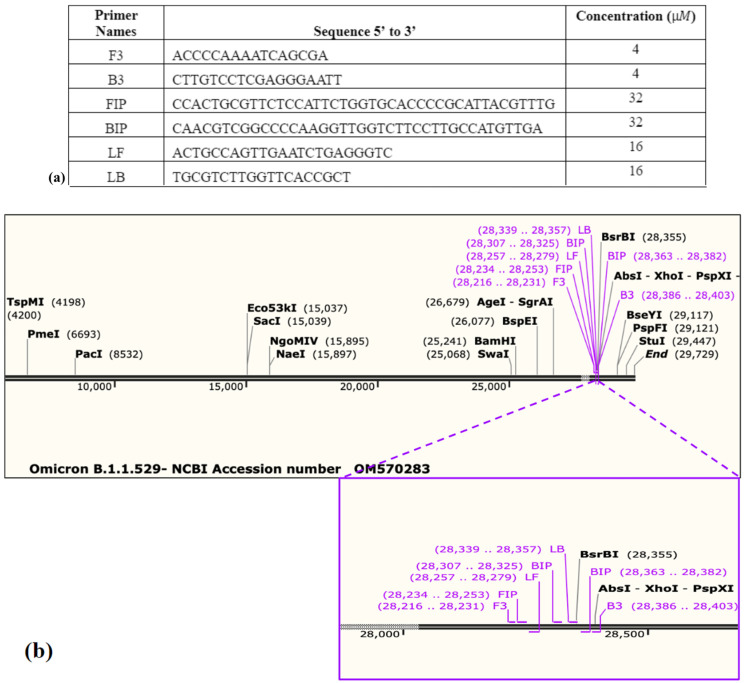
(**a**) SARS-CoV-2 RT-LAMP primer sequences alongside labeled sequences and their concentration. (**b**) The SARS-CoV-2 RT-LAMP primer aligned against the Omicron-B.1.1.529 variant (GenBank: OM570283.1) in the software SnapGene (https://www.snapgene.com/) illustrating the primers’ annealing sites on the genome.

**Figure 4 viruses-16-01190-f004:**
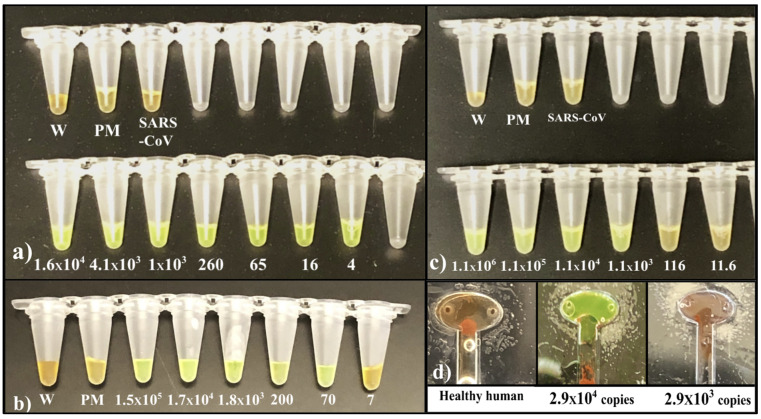
(**a**) The RT-LAMP SARS-CoV-2 primer analytical sensitivity and specificity colorimetric test using SARS-CoV-2 RNA (positive control) and SARS-CoV RNA (negative control). Color change from orange to green was observed following the amplification in the reactions using SYBR Green I dye. The numbers indicate target copies/reaction. (**b**) The RT-LAMP colorimetric detection test of SARS-CoV-2 from 100 μL of saliva spiked with the SARS-CoV-2 target plasmid. The number of SARS-CoV-2 RNA target copies per reaction utilized in the assay is indicated. (**c**) RT-LAMP colorimetric detection test results of the saliva spiked with inactivated virions. The number of viral RNA copies per reaction is indicated. (**d**) An i mage of the reaction chamber after the fully automated chip run of the inactivated virus-spiked saliva samples. The number of viral RNA copies loaded per assay is indicated. (Abbreviations: W—Water.; PM—Primer + MasterMix).

**Figure 5 viruses-16-01190-f005:**
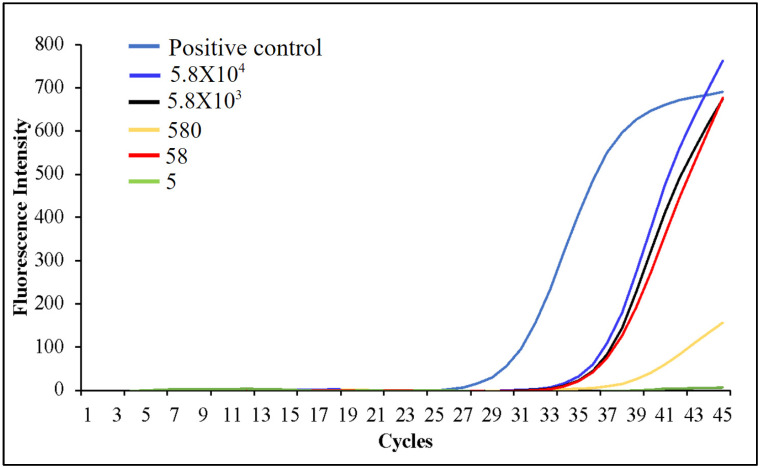
Amplification curves from the SARS-CoV-2 benchtop RT-qPCR assay on the serially diluted saliva- spiked samples. The assay can detect as low as 58 SARS-CoV-2 RNA copies/reaction, validating that the Dynabeads Silane viral NA kit works well not only with serum/plasma samples but also with saliva samples.

**Figure 6 viruses-16-01190-f006:**
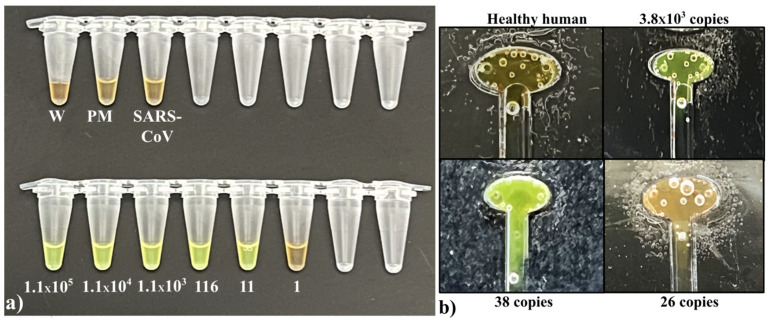
(**a**) Benchtop RT-LAMP colorimetric detection test results of the nasopharyngeal samples spiked with inactivated virus samples. The number of viral RNA copies per reaction is indicated. (**b**) An i mage of the reaction chamber after the fully automated chip run of the nasopharyngeal samples spiked with inactivated virus samples. The number of viral RNA copies per assay utilized is indicated. (Abbreviations: W—Water; PM—Primer + Maste.rMix).

## Data Availability

All data needed to evaluate the conclusions are present in the paper and/or the Appendix A. Additional data related to this paper may be requested from the authors.

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
