# Peer review of "Development of a Diagnostic Microfluidic Chip for SARS-CoV-2 Detection in Saliva and Nasopharyngeal Samples"

_viruses, 2024, doi:10.3390/v16081190_

Round 1
Reviewer 1 Report
Comments and Suggestions for Authors
In their Manuscript “Development of a Diagnostic Microfluidic Chip for SARS-CoV-2 Detection in Saliva and Nasopharyngeal Samples”, Sharma et al. describe their efforts to establish a semi-automatic point-of-care test for the detection of SARS-CoV-2 RNA from samples acquired from the upper respiratory tract of patients using loop-mediated isothermal amplification (LAMP) rtPCR. For their assay the authors designed primes targeting a conserved region of the SARS-CoV-2 nucleocapsid gene which they identified from 75 SARS-CoV-2 sequences. In the assay, RNA from lysed samples is bound to magnetic beads, which move through different chambers of a microfluidics chip for washing, reverse transcription and amplification via LAMP PCR. Successful amplification of viral nucleic acids was evaluated by naked eye after addition of SYBR green dye. Saliva and nasopharyngeal swab material spiked with viral RNA or heat-inactivated virions were utilized to assess the assay’s analytical sensitivity, which was found to be comparable to that of the gold standard assay, i.e. RT-qPCR.
Because of their comparably low diagnostic sensitivity and specificity, rapid antigen tests (RATs) are often considered inadequate for point-of-care COVID-19 testing. Thus, there is a demand for novel assays to overcome the shortcomings of RATs. The assay described in this manuscript, could become an advantageous alternative for point-of-care testing. This study could, therefore, be of interested to the readership of Viruses. However, several adjustments need to be made to make this manuscript eligible for publication in the journal.
Specific comments:
Albeit there being different experimental setups, LAMP PCR on microfluidic chips is an often-published method, even for the detection of SARS-CoV-2 RNA. The assay described in this manuscript combines the use of magnetic beads for sample processing and an assay readout by naked eye. Both could be beneficial for point-of-care testing. However, to make this manuscript stand out, the performance of the herein described assay for the detection of acute COVID-19 compared to RT-qPCR should be investigated in greater detail.
For assessing the analytical sensitivity of the assay, inactivated virions and genomic RNA from an ancestral SARS-CoV-2 strain were used. I highly recommend investigating the diagnostic performance of the assay utilizing swab and/or saliva samples from COVID-19 patients. Using a set of samples acquired from the upper respiratory tract of individuals infected with SARS-CoV-2, the sensitivity of the assay compared to RT-qPCR could be measured. Similarly, the specificity of the LAMP-PCR on the microfluidic chip could be tested using samples from healthy individuals. Such samples from COVID-19 patients and healthy individuals may be available in large quantities from diagnostic laboratories or biobanks.
The primers described in this study were designed using a dataset of 75 SARS-CoV-2 sequences. Please indicate how these 75 SARS-CoV-2 sequences were chosen, e.g., whether these sequences were selected from a divers set of sequences from different of SARS-CoV-2 variants. In the last, 4 years SARS-CoV-2 underwent considerable evolution and genetically diverse strains emerged that show alterations also in the nucleocapsid gene. Using a computational approach and a larger dataset, the authors should check if their primer can bind to the nucleocapsid gene in a greater variety of SARS-CoV-2 strains, including those from the different variants of concern and their subvariants.
If I understand correctly, the authors did not only evaluate their results by the naked eye but also took images of their reaction chambers after the addition of SYBR green and analyzed them using the software FIJI (see supplementary figures S6 and S7). Please add additional information describing in greater detail how this image analysis was performed.
Several figures, including figures 4 and 5, use abbreviations that are not explained in the figure legends. Please explain abbreviations in the figure legends. Similarly, some terminology used in the figures, e.g., in Figure 1, is dissimilar to the terminology used in the text. Please unify your terminology.
I was unable to access the supplementary video file. Please make the video file accessible for the reviewers.
Figure 3 consists of a table and two screenshots obtained from the software SnapGene, all in low resolution. In the figure there is a mouse icon in the upper right corner that assumingly does not belong there. In my opinion, the depiction of the primer sequences and their alignment to an example for a SARS-CoV-2 genome should be re-organized and shown in with more clarity.
Reviewer 2 Report
Comments and Suggestions for Authors
The manuscript by Sharma describes a microfluidic chip that can be used for a rapid test for the point of care detection of SARS CoV-2 RNA. The RT-LAMP technology, which allows the detection of nucleic acids without the use of a thermocycler, has already been developed for this and other viruses with equal or perhaps superior sensitivity (as low as 2 copies it is claimed) and as indicated in the introduction, some are already available commercially. The authors propose that the novelty of this manuscript is the chip that makes the reactions easier to perform and the readout straightforward.
Overall, this is an interesting manuscript, and this chip may prove to be an advancement, though its sensitivity is less than others already described (Huang et al. https://doi.org/10.1016/j.ebiom.2021.103736, and others). The manuscript is well-written, and the figures are illustrative. I have the following comments and suggestions for the authors.
1. The key figure S5 that is used to determine the sensitivity of the assay should be added to the manuscript. Furthermore, the graph lines are unclear. Why does the yellow line (580 copies) demonstrate less fluorescence intensity than the red line (58 copies)?
2. Did the authors ever test the chip system with actual clinical samples rather than saliva that had been spiked with RNA (recognizing that the limit of detection determination requires the spiking experiment)?
3. In Fig S6, how was the ‘arbitrary’ threshold set?
4. Did the authors test their chip side-by-side with a commercial RT-LAMP system, for example the NE Biolabs?
5. Since the novelty is the microchip and its process, the authors should provide a permanent video record (I was unable to view one) for those interested in the process.
6. The authors should refer to the negative viral control as either SARS CoV or preferably SARS CoV-1.
Round 2
Reviewer 1 Report
Comments and Suggestions for Authors
Thank you for revising the manuscript. Many of my concerns have been addressed. However, I still have several suggestions to improve the paper:
I still believe that the significance of the paper could be greatly increased by addressing the diagnostic sensitivity and specificity of the presented assay utilizing a set of saliva and/or nasopharyngeal swab samples from healthy individuals and COVID-19 patients.
Regarding the 75 sequences that were selected for the primer design, please indicate on what rationale these sequences were chosen. The authors show that their primers bind to SARS-CoV-2 variant Omicron B.1.1.529. In my opinion however, it would be beneficial to use in silico analyses and investigate the conservation of the primer binding regions across a wide variety of SARS-CoV-2 variants. In their response to the reviewer, the authors indicate that the primers of the assay can be re-designed and adapted, in case of mutations in the primer binding regions due to genetic drift of the viral genome. Even though such a statement might be insignificant and easy to comprehend for most readers, I still recommend including it in the Results & Discussion section of the manuscript.
When explaining the FIJI analysis of the results in greater detail, the authors introduce the measurement of arbitrary units (a.u.). Please elaborate on this unit. Is “a.u.” a unit for the grey intensity that is defined by FIJI or did the authors themselves define the unit? If the authors introduce arbitrary units by themselves, please indicate how this unit was defined. On what mathematical basis was the threshold for positivity (i.e. 20 a.u.) chosen?
When elaborating on the analytical sensitivity their assay, the authors claim that the limit of detection of the test equals to a certain number of “virion copies”. As the assay was validated against RT-qPCR, it would be, in my opinion, more accurate to use the term “viral RNA copies” instead of “virion copies”.
In order to make the results of the analytical sensitivity more comparable to other research, I highly recommend to show the results not as viral RNA copies/reaction but instead as viral RNA copies/mL of the input sample (e.g., swab material, saliva).
The analytical sensitivity of the developed LAMP assay in saliva samples was approximately 1000 viral RNA copies/reaction, when performed in the test tube, and 30,000 copies/reaction, when performed on the microfluidics chip. The sensitivity of the RT-qPCR in saliva samples, however, was roughly 60 viral RNA copies/reaction. The authors claim that their assay’s sensitivities are “comparable” to that of the RT-qPCR, albeit there being a >10x and >100x difference, respectively. Please re-phrase your statement emphasizing the still superior sensitivity of the RT-qPCR.
In the manuscript the term “limit of detection” is abbreviated with “LoD” in some cases, and in others the abbreviation “LOD” is used. Please unitize the abbreviation.
In the manuscript the unit for volume is often written as “𝜇LmL”. I believe this is an error and the others meant to write “𝜇L” instead. Please correct this typo accordingly.
In Figure 4, negative controls are labeled with SAR-CoV. I do believe that they should be labeled with SARS-CoV. Please correct the labeling.
In figure legend 4b the authors state that “saliva spiked with the SARS-CoV-2 target RNA” was used for the assay shown in the figure subpanel. However, in the manuscript text the authors write that they spiked plasmid into the saliva for this analysis. Please clarify this discrepancy and correct the text if applicable.
Figure legend 5 should be expanded. It should describe the positive control in greater detail and the numbers below the positive control (assumingly the input amount of inactivated virus). Please discuss the rather surprising result of more PCR cycles being necessary for 580 inactivated virus copies to become positive than for 58.
Reviewer 2 Report
Comments and Suggestions for Authors
The manuscript has been improved. I am still concerned about the odd curve in Fig 5 (now in the manuscript). The explanation that the RNA was degraded may well be correct, but why only in this sample, and why wasn't it then repeated to give a proper curve.
The manuscript would reach a higher level of enthusiasm if the clinical samples were included. I'll leave it to the authors whether they want to enhance the manuscript that way.
